# Indicators of the Right Ventricle Systolic and Diastolic Function 18 Months after Coronary Bypass Surgery

**DOI:** 10.3390/jcm11143994

**Published:** 2022-07-10

**Authors:** Alexey N. Sumin, Anna V. Shcheglova, Ekaterina V. Korok, Tatjana Ju. Sergeeva

**Affiliations:** Laboratory of Comorbidity in Cardiovascular Diseases, Department of Clinical Cardiology, Federal State Budgetary Scientific Institution “Research Institute for Complex Issues of Cardiovascular Disease”, Sosnoviy Blvd., 6, 650002 Kemerovo, Russia; karpav@kemcardio.ru (A.V.S.); korok-82@mail.ru (E.V.K.); sergeeva.tatiana65@yandex.ru (T.J.S.)

**Keywords:** coronary artery disease, coronary artery bypass surgery, right ventricle, systolic function, diastolic function, postoperative echocardiography

## Abstract

Objective. Right ventricular (RV) dysfunction after coronary artery bypass grafting (CABG) is associated with increased mortality and morbidity. In previous studies, the parameters of RV systolic function were mainly assessed, while the dynamics of RV diastolic function after surgery was practically not studied. The aim of this study was to study the dynamics of indicators of systolic and diastolic RV function after CABG as well as to identify factors associated with their presence. Methods. The study included 160 patients who underwent CABG and 36 volunteers with no history of coronary artery disease (CAD) as a control group. Echocardiographic examination of patients was performed to assess systolic and diastolic RV dysfunction before surgery and 18 months after CABG. A level of s’t < 10 cm/sec or TAPSE < 16 mm was considered as a sign of existing RV systolic dysfunction. RV diastolic dysfunction was defined as an Et/At ratio < 0.8 or >2.1 and/or an Et/et’ ratio > 6. Results. In CAD patients 18 months after CABG, there was an increase in the frequency of the right ventricular systolic (from 7.5% to 30%, *p* < 0.001) and diastolic (from 41.8% to 57.5%, *p* < 0.001) dysfunction. An increase in TAPSE (*p* = 0.007), a decrease in e’t (*p* = 0.005), and the presence of RV systolic dysfunction before surgery (*p* = 0.023) was associated with a significant increase in the likelihood of detecting RV systolic dysfunction 18 months after CABG (χ^2^(3) = 17.4, *p* = 0.001). High values of At before surgery (*p* = 0.021) and old myocardial infarction (*p* = 0.023) were significantly associated with an increased likelihood of detection of RV diastolic dysfunction 18 months after CABG (χ^2^(2) = 10.78, *p* = 0.005). Conclusions. This study demonstrated that in CAD patients 18 months after CABG, there was an increase in the frequency of right ventricular systolic and diastolic dysfunction. We also established the initial clinical, echocardiographic parameters, and perioperative complications associated with the presence of these changes in the postoperative period. The clinical and prognostic significance of the presence of systolic and/or diastolic RV dysfunction in patients 18 months after CABG remains to be explored.

## 1. Introduction

Myocardial revascularization is an effective method for improving the prognosis of patients with coronary artery disease (CAD). In the presence of multivessel disease or systolic dysfunction, coronary artery bypass grafting (CABG) is indicated in such patients. [1]. However, due to the traumatic nature of the operation, patients may develop significant postoperative complications that can worsen the prognosis. For example, the development of right ventricular failure after cardiac surgery is associated with a high complication rate and poor prognosis. [2,3,4,5]. Accordingly, the diagnosis and treatment of RV dysfunction after CABG are important for prognosis improvement, both in the early postoperative period and in the more distant periods [6,7,8,9]. It should be noted that in previous studies, the parameters of the RV systolic function were mainly assessed [10,11,12,13]. At the same time, there is evidence that RV diastolic dysfunction can evolve before the identification of RV systolic dysfunction [14,15] and is associated with the development of perioperative complications [16]. While there is information regarding the indicators of RV systolic function in the postoperative period of cardiac surgery [17,18], the dynamics of RV diastolic function after surgery has not been practically studied. The aim of this study was to study the midterm dynamics of the indicators of systolic and diastolic RV function after CABG as well as to identify factors associated with their presence.

## 2. Subjects, Materials, and Methods

### 2.1. Study Population

The study included 200 patients who underwent CABG in the cardiovascular surgery department at our institute. A description of the recruitment and selection of patients in the study has been provided in our previous articles [15,16], and it is also presented in Figure 1. In patients, we assessed preoperative parameters: risk factors, the presence of cardiovascular diseases, comorbidities, the severity of coronary and heart failure, damage to extracoronary arteries, laboratory data, and therapy received. According to an extended protocol, we evaluated echocardiography with the study of both systolic and diastolic parameters of the left and right ventricles. Follow up was performed after 18 months by an outpatient visit with 166 (83%) patients, of which 160 participants in the study underwent repeated echocardiography with an assessment of the right ventricular systolic and diastolic function (Figure 1).

Additionally, the echocardiographic parameters were evaluated in the control group, which consisted of 36 volunteers with no history of coronary artery disease and comparable in age and gender with the study sample of patients.

### 2.2. Data Collection and Study Outcomes

In the present study, we evaluated the dynamics of transthoracic echocardiography parameters in CAD patients 18 months after CABG compared with preoperative data. Detailed patient characteristics were presented in a previously published article [15]. We also recorded a number of preoperative and intraoperative parameters (i.e., results of coronary angiography, intraoperative characteristics, and combined operations). The primary outcome measure was the comparative frequency of RV systolic and diastolic dysfunction (RVSD and RVDD) before and 18 months after CABG in CAD patients. Then, we studied the dynamics of the left and right ventricles parameters before and after CABG in comparison with the data of the control group. We also assessed the factors (i.e., initial data, intraoperative parameters, and postoperative complications) associated with the RVSD and RVDD presence after CABG using logistic regression analysis. Additionally, we used multiple linear regression analysis for assessment the relationship between indicators of RV systolic and diastolic function 18 months after CABG with clinical and anamnestic variables.

### 2.3. Echocardiographic Examination

Standard transthoracic echocardiography was performed on a Philips Clear Vue 550 ultrasound scanner (USA) according to current recommendations [19]. The detailed protocol and examination methodology were presented earlier [15]. The analysis of the left heart included an assessment of the size and volume parameters of the left ventricle, stroke ejection, LV myocardial mass, diameter of the left atrium (LA), and left ventricular ejection fraction (LVEF). When analyzing the right heart, the following were assessed: the size of the right atrium (RA) and RV, the wall thickness of the right ventricle in diastole (RVD), and tricuspid annular plane systolic excursion (TAPSE).

In the Doppler mode, the LV functional parameters in diastole were studied: the peak velocity of early (E) and late transmitral filling (A) and their ratio (E/A), isovolumetric relaxation time (IVRT). In a similar way, the RV diastolic function was assessed: the velocity of early (Et) and late (At) transtricuspid RV filling and their ratio (Et/At).

In the spectral tissue Doppler mode, we assessed the velocity of early and late diastolic movement of the mitral and tricuspid valve annulus and their ratio (e’, a’, e’/a’, E/e’; e’t, a’t, e’t/a’t, Et/e’t), the velocity of the systolic movement of the mitral (s’) and tricuspid (s’t) valve annulus. We also evaluated the Tei index for both ventricles.

As a criteria for RV systolic dysfunction, we used the following indicators: s’t level < 10 cm/sec or TAPSE < 16 mm. Et/At ratio values < 0.8 or >2.1 as well as and/or an Et/et’ ratio > 6 were signs of diastolic RV dysfunction [20].

### 2.4. Ethics Statement

The study was conducted in accordance with the Helsinki Declaration; all patients signed informed consent to the study. The study protocol was approved by the Local Ethics Committee of the FSBSI Research Institute for Complex Issues of Cardiovascular Diseases.

### 2.5. Statistical Analyses

Statistical data processing was performed using the standard software packages Statistica 8.0 and SPSS 17.0. The distribution of quantitative data was checked using the Shapiro–Wilk test. Given that the distribution of all quantitative characteristics differed from normal, they are presented as the median and quartiles (25th and 75th percentiles). The groups were compared using the Kruskal–Wallace, Mann–Whitney, and χ^2^ tests. To solve the problem of multiple comparisons, Bonferroni correction was used.

To identify factors associated with the presence of RVSD and RVDD 18 months after CABG, we used binary logistic regression. We included in the model for evaluation both demographic parameters (i.e., gender and age) and baseline clinical and laboratory parameters, presence of perioperative complications, preoperative echocardiographic parameters (i.e., LA, LVEF, E/A ratio, E/e’, s’, index LV Tei, TAPSE, index RV Tei, Et, At, ET/At, e’t, a’t, e’t/a’t, s’t, and Et/e’t). Additionally, we assessed the relationship between individual measures of right ventricular systolic and diastolic function with clinical and history variables 18 months after CABG using stepwise multiple linear regression analysis. The level of critical significance (*p*) during the regression analysis was taken equal to 0.05.

## 3. Results

The mean age of the patients was 63.5 ± 6.4 (from 34 to 80) years, and the mean follow-up period was 17.7 ± 0.7 months. There was a significant increase in the number of patients with RVSD (from 7.5% to 30.0%, *p* < 0.001) and RVDD (from 41.8% to 57.5%, *p* < 0.001) 18 months after CABG compared with initial data (Figure 2).

The initial characteristics of the patients in the preoperative period are presented in Table 1. Most of the patients had arterial hypertension (94.3%), 26.8% patients had type 2 diabetes, and the average body mass index (BMI) was 28.7 ± 3.7. The smokers comprised 30.2% of patients, and dyslipidemia occurred in 65.6% of patients. A history of myocardial infarction was observed in 62.3% of patients. Percutaneous coronary intervention was previously performed in 17.6% of patients. Carotid artery stenosis > 50% was detected in 19.4% patients, and 8.2% patients had a history of stroke. Stenosis of the lower extremity arteries >50% was observed in 4.4% of patients. Rhythm disturbance occurred in 16.3% of cases. All patients before and after CABG received optimal drug therapy. In the early postoperative period, 8.1% of patients had postoperative heart failure, 1.3% had nonfatal myocardial infarction, and 3.1% had stroke.

Assessment of the structural and functional parameters of the left heart showed (Table 2) that in the CAD group, in relation to the control group, the end-systolic size and end-systolic volume were higher (*p* < 0.001). There was also a significantly larger LA size in the CAD group, mainly after CABG, in comparison with the control group (*p* < 0.001). The median LV ejection fraction in all groups was within the standard values; however, it was the smallest in the CAD group 18 months after CABG (*p* < 0.001). In addition, an increase in e’ and a decrease in a’ were noted in patients of the CAD group after CABG (*p* < 0.001). Accordingly, the ratio e’/a’ was significantly higher in patients after CABG (*p* < 0.001). At the same time, the LV Tei index was higher in the CAD group of patients before CABG (*p* < 0.001). The ratios of E/e’ in all groups were within the normative values; however, the ratio of E/e’ in patients of the CAD group before CABG was higher compared to the group after CABG (*p* < 0.001). Moreover, in the CAD group, lower s’ were found in comparison with the control group (*p* < 0.001).

The RV systolic function was significantly lower in the CAD group after CABG (Table 3); this group had the lowest values of RVEF (*p* < 0.001), s’t (*p* < 0.001), and TAPSE (*p* < 0.001). Similar differences were also found for the parameters of RV diastolic function.

The level of pressure in the pulmonary artery was significantly higher in the CAD group compared to the control group (*p* < 0.001). The assessment of transtricuspid flows revealed differences in the velocities of early diastolic transtricuspid flow (Et) and late diastolic transtricuspid flow (At) in the CAD group compared with the control group (*p* < 0.001). The ratio of early and late diastolic transmitral flows was significantly higher in the CAD patients before CABG (*p* < 0.001) compared with the control).

Significant differences were demonstrated in Doppler imaging tissue of the tricuspid valve annulus. There was a decrease in e’t and a’t in the CAD group, mainly after CABG compared to the control (*p* < 0.001). Accordingly, the e’t/a’t ratio was significantly higher in CAD patients after CABG compared to the baseline and control group (*p* < 0.001). There was a decrease in s’t after CABG relative to the control and CAD patients before CABG (*p* < 0.001). The Et/e’t ratio in the groups also had significant differences and was the highest in patients over time after CABG (*p* < 0.001). The values of the Tei RV index in patients after CABG also significantly decreased (*p* < 0.0001).

The dynamics of the indicators of systolic (i.e., s’t and TAPSE) and diastolic (i.e., Et/At and Et/e’t) RV functions before and after CABG are shown in Figure 3.

Binary logistic regression analysis was performed to identify the factors associated with the presence of RVSD and RVDD 18 months after CABG (Table 4 and Table 5). An increase in TAPSE, a decrease in e’t, and the presence of RV systolic dysfunction before surgery was associated with a significant increase in the likelihood of detecting RV systolic dysfunction 18 months after CABG (Table 4). The logistic regression model was statistically significant, χ^2^(3) = 17.4, *p* = 0.001. The model explained 14.7% (Nagelkerke R2) of the variance in RVSD presence and correctly classified 68.4% of the cases. Increasing At before surgery and myocardial infarction history were significantly (χ^2^(2) = 10.8, *p* = 0.005) associated with an increased likelihood of detection of RV diastolic dysfunction 18 months after CABG (Table 5). 

A stepwise multiple linear regression was conducted to predict parameters of RV systolic (s’t) and diastolic (Et/e’t) function 18 months after CABG. The models included age, gender, history of myocardial infarction, echocardiography parameters before surgery (i.e., LA, LVEF, E/A ratio, E/e’, s’, LV index Tei, TAPSE, RV index Tei, Et, At, ET/At, e’t, a’t, e’t/a’t, s’t, and Et/e’t), biochemical parameters (i.e., glucose, cholesterol, and creatinine), and perioperative complications (i.e., myocardial infarction, stroke, renal failure, and heart failure). The variables e’t and s’t before surgery statistically significantly predicted s’t after CABG (Table 6): F(2, 155) = 8.09, *p* < 0.001, R2 = 0.096. In addition, age, LA dimensions, Et before surgery, and perioperative stroke significantly predicted ratio Et/e’t 18 months after CABG (Table 7): F(4, 155) = 5.78, *p* < 0.001, R2 = 0.133.

## 4. Discussion

The present study shows that in CAD patients 18 months after CABG, there was a deterioration in both systolic and diastolic right ventricular function. Moreover, if before the operation, systolic dysfunction was rarely detected, then the operation led to its more frequent detection. Independent factors associated with the detection of RV systolic dysfunction after CABG were TAPSE, e’t, and RV systolic dysfunction before surgery and with the detection of RV diastolic function—the previous myocardial infarction and values At before surgery.

Decreased RV function after cardiac surgery may persist for up to a year after surgery and often results in incomplete recovery on echocardiographic monitoring. For example, in the work of Chinikar M et al. [18], in patients with normal RV function before surgery, a week after CABG, disturbance of RV systolic function occurred in more than half of patients (abnormal TAPSE in 81.0% of patients, Tei index in 79.0%, and s’t in 62.0%). After 6 months, the disturbance frequency decreased but remained significant—abnormal TAPSE was detected in 49.0% of cases, Tei in dexin 49.0%, and s’t in 37.0%). A year after CABG, a significant decrease in the RV systolic function was revealed compared to preoperative values (TAPSE from 21.7 to 12.1 mm, s’t from 14.0 to 7.0 cm/s) [21]. Our study showed that RV systolic dysfunction persisted up to 1.5 years in 30% of patients. The development of such dysfunction can be influenced by various perioperative factors—sternotomy, pericardial incision, and cardioplegia variants. A recent study showed that a decrease in the mechanical function of the right ventricle (as assessed using a three-dimensional-derived RV free-wall strain) does not occur during surgery (before and after sternotomy) but after isolated CABG with cardiopulmonary bypass [22]. However, a significant decrease in right ventricular longitudinal function (TAPSE and s’) was also noted after off-pump CABG [23]. The use of mitral valve repair through a minimally invasive surgery approach (with lateral pericardial incision) was accompanied by a smaller decrease in TAPSE in the postoperative period than in traditional surgery with anterior pericardial incision [17]. On the other hand, diminished RV function proved to be independent of the number of grafts and right coronary artery revascularization [21]. In addition, after thoracic surgery, there were no changes in diastolic or systolic RV function, which were observed after heart surgery with pericardiotomy [24]. In patients with a mild reduced preoperative RV systolic function, a significant impairment of the contractile apparatus was observed when it was assessed using an experimental model of right atrial fibers [25]. Therefore, it is not surprising that this disturbance increased in the postoperative period. 

Previous studies have shown that in patients with CAD before CABG, predictors of mild RV systolic dysfunction, in addition to low LV ejection fraction [25,26], were also carbohydrate metabolism disorders, arrhythmias, an increased risk of surgery according to the EuroScore II scale, as well as a decrease in kidney function [25]. At the same time, the localization and number of coronary stenoses and the localization of a previous myocardial infarction did not affect the frequency of detection of the right heart dysfunction [26]. For RV diastolic dysfunction, an association with a decrease in LV systolic function was also noted and, additionally, with the age of patients, a history of myocardial infarction, multifocal atherosclerosis, and impaired lipid metabolism [15]. In the present study, a history of myocardial infarction was also associated with RV diastolic dysfunction after CABG, along with one of the indicators of RV diastolic function before surgery (At).

A decrease in RV ejection fraction before CABG is associated with a more complicated postoperative course and length of stay in the intensive care unit [11], a greater number of perioperative complications, and long-term mortality after CABG [4], as well as a higher rate of rehospitalization after surgery [27]. A decrease in preoperative diastolic RV function is one of the independent risk factors for early death after CABG in patients with severe left ventricular dysfunction [28]. In addition, even though mild reduced right heart function (TAPSE in the normal range but less than 20 mm) had no impact on 30 day clinical outcome parameters, it was associated with a significantly longer hospital stay [25]. Moreover, in patients with baseline normal RV systolic function, the presence of RV diastolic dysfunction is associated with a higher incidence of postoperative heart failure requiring prolonged inotropic therapy [16]. There are fewer data on the clinical and prognostic significance of the development of RV dysfunction in the postoperative period. For example, the development of severe RV insufficiency after cardiac surgery was accompanied by severe complications and adverse outcomes [5]. However, the frequency of such significant right ventricular dysfunction in that study was low; thus, the less pronounced manifestations of right ventricular dysfunction detected by echocardiography attracted the attention of investigators. The presence of RV systolic dysfunction 6 months after CABG was associated with a decrease in the functional state of patients (decrease in distance during the six-minute walk test) [18]. On the other hand, despite the decrease in RV function, the functional class in patients improved 1 year after CABG [21]. The clinical significance of reduced systolic and/or diastolic RV function 18 months after CABG remains to be explored.

The present study had a number of limitations; therefore, its results should be interpreted with caution. First, it was single-center, which does not allow for the distribution of its results to other centers. Secondly, only a Caucasian population was surveyed; thus, its results cannot be extended to other ethnic groups. Thirdly, the study did not include patients with comorbidities, which reduces the generalizability of the results obtained, primarily to patients with lung diseases and valvular pathology. Moreover, the design of the study did not include an assessment of right ventricular function in the early postoperative period. Therefore, our data do not allow for elucidating the proportion of patients with RVSD or RVDD in the early postoperative period and comparing the rates of early postoperative and medium-term RV dysfunction. In addition, when determining the RV function, we did not use such modern methods of assessment as RV speckle-tracking measurement and RV 3D echocardiography [29]. However, we relied on data from real clinical practice, which shows that methods such as RV global longitudinal deformation and RV 3D echocardiography are used extremely rarely (3% and 1%, respectively) [30].

## 5. Conclusions

In CAD patients 18 months after CABG, there was an increase in the frequency of right ventricular systolic (from 7.5% to 30%) and diastolic (from 41.8% to 57.5%) dysfunction. The presence of RV systolic dysfunction after CABG was associated with TAPSE, e’t, and RV systolic dysfunction before surgery. The detection of RV diastolic function after CABG was associated with the previous myocardial infarction and values of At before surgery. The clinical and prognostic significance of the presence of systolic and/or diastolic RV dysfunction in patients 18 months after CABG remains to be explored.

## Figures and Tables

**Figure 1 jcm-11-03994-f001:**
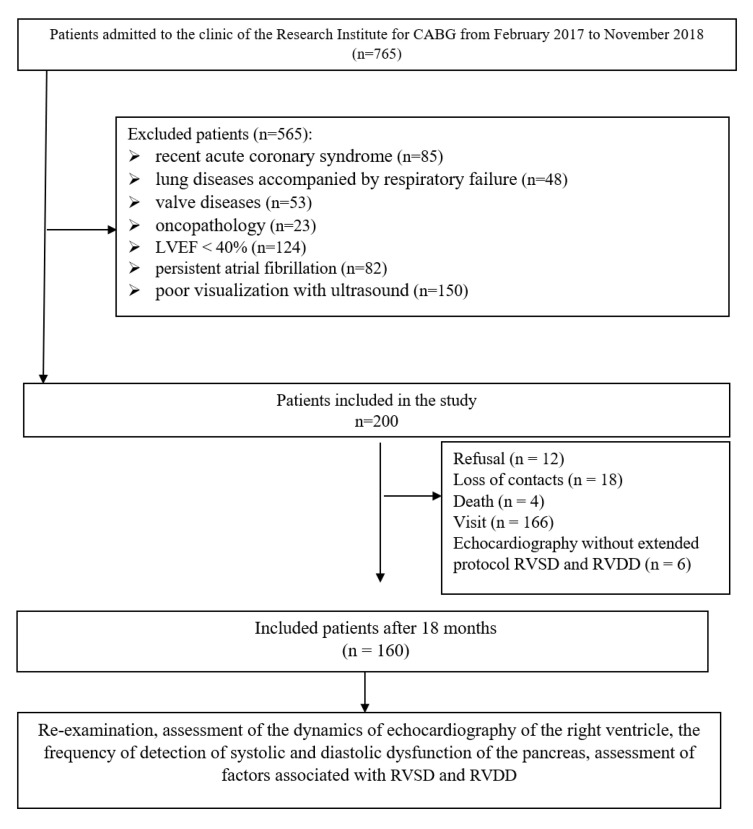
Study flowchart.

**Figure 2 jcm-11-03994-f002:**
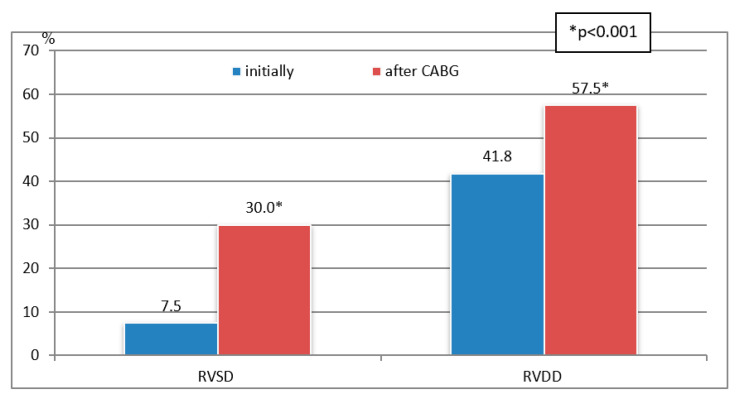
Frequency of the RVSD and RVDD before and after CABG.

**Figure 3 jcm-11-03994-f003:**
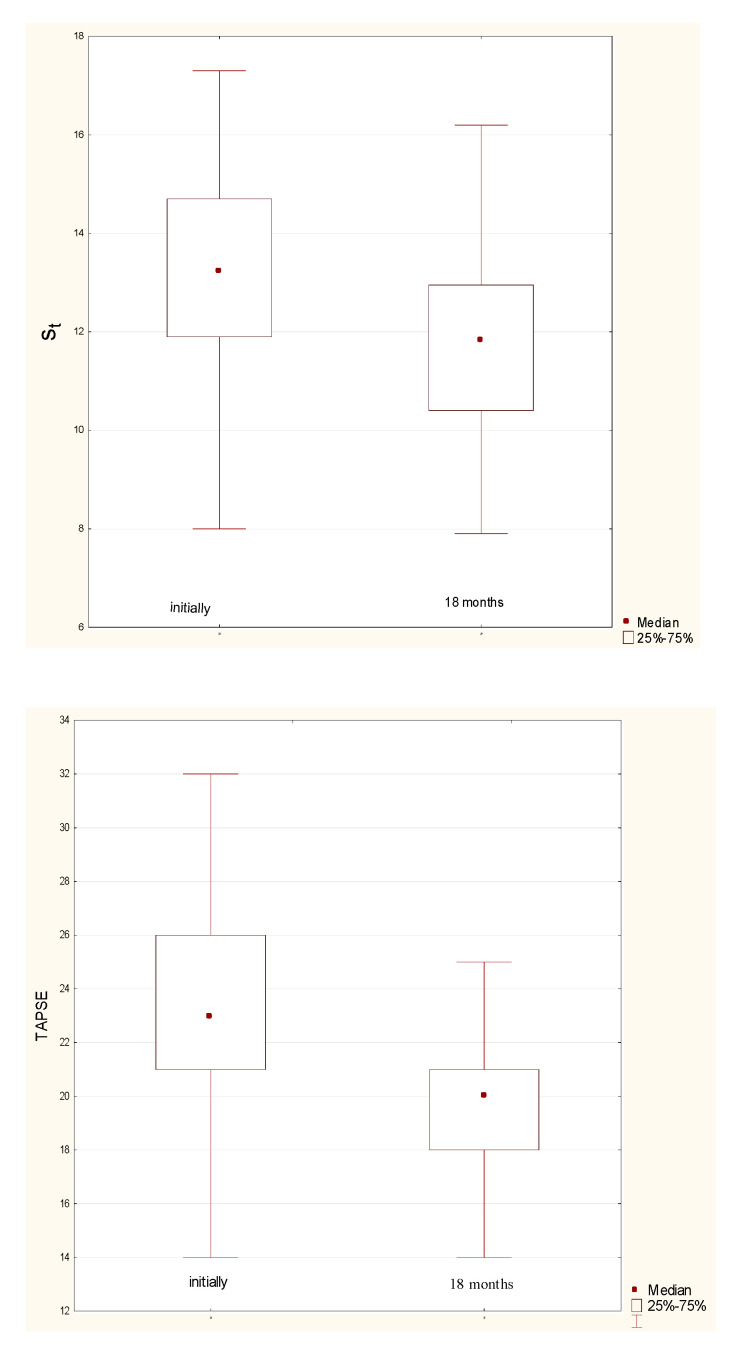
Dynamics of indicators of systolic (i.e., s’t and TAPSE) and diastolic (i.e., Et/At and Et/e’t) RV function before CABG and after 18 months.

**Table 1 jcm-11-03994-t001:** General characteristics of patients.

Characteristics	*p*
Clinical and anamnestic data	
Male, *n* (%)	120 (75.0)
Age, ME [LQ; UQ], (y)	64.0 [60.0; 68.0]
Body mass index, ME [LQ; UQ], (kg/m^2^)	28.7 [26.0; 30.7]
Smoking, *n* (%)	48 (30.2)
Smoking experience, ME [LQ; UQ], (y)	40.0 [20.0; 40.0]
Hypertension, *n* (%)	151 (94.3)
Angina pectoris, *n* (%)	138 (86.3)
Myocardial infarction history, *n* (%)	99 (62.3)
CHF NYHA IIA cт., *n* (%)	31(19.4)
Rhythm disturbances, *n* (%)	26 (16.3)
Diabetes mellitus, *n* (%)	43 (26.8)
Stroke history, *n* (%)	13 (8.2)
PCI history, *n* (%)	28 (17.6)
Chronic lung disease, *n* (%)	25 (15.6)
Carotid atherosclerosis >50%, *n* (%)	31 (19.4)
Atherosclerosis of the legs >50%, *n* (%)	7 (4.4)
Hyperlipidemia, *n* (%)	105 (65.6)
Right coronary artery stenosis >70%, *n* (%)	141(88.1)
Biochemical markers	
NT-proBNP, ME [LQ; UQ], pg/mL	66.7 [32.4; 150.0]
Creatinine, ME [LQ; UQ], mmol/L	84.5 [74.5; 97.0]
Glucose, ME [LQ; UQ], mmol/L	5.7 [5.2; 6.5]
Total cholesterol, ME [LQ; UQ], mmol/L	4.6 [3.7; 5.3]
Low-density lipoprotein, ME [LQ; UQ], mmol/L	2.73 [2.08; 3.54]
High-density lipoproteins, ME [LQ; UQ], mmol/L	1.13 [0.92; 1.33]
Triglycerides, ME [LQ; UQ], mmol/L	1.43 [1.16; 2.17]
Therapy before surgery	
Betablocker, *n* (%)	152 (95.0)
Statins, *n* (%)	152 (95.0)
ACE-I, *n* (%)	121 (75.6)
Aspirin, *n* (%)	148 (92.5)
Surgical procedure	
Cardiopulmonary bypass, *n* (%)	184 (92.0)
Bypass graft number	3 (2–3)
Cardiopulmonary bypass duration, ME [LQ; UQ] (min)	77.0 [65.0; 95.0]
Aortic cross-clamp time, ME [LQ; UQ] (min)	51.0 [40.0; 61.0]
Ventriculoplasty, *n* (%)	17 (8.5)
Thrombectomy, *n* (%)	9 (4.5)
Radiofrequency ablation, *n* (%)	6 (3.0)
Carotid endarterectomy, *n* (%)	23 (11.5)
Perioperative complications	
Congestive heart failure, *n* (%)	13 (8.1)
Myocardial infarction, nonfatal, *n* (%)	2 (1.3)
Stroke, nonfatal, *n* (%)	5 (3.1)
Cardiovascular death, *n* (%)	-
Acute renal failure, *n* (%)	5 (3.1)
Therapy after surgery	
Betablocker, *n* (%)	139 (86.9)
Statins, *n* (%)	137 (85.6)
ACE-I, *n* (%)	96 (60.0)
Aspirin, *n* (%)	130 (81.3)

Continuous data are presented as the median [lower quartile, upper quartile]; CHF—chronic heart failure; NYHA—New York Heart Association; PCI—percutaneous coronary intervention; NT-proBNP—N-terminal pro B-type natriuretic peptide; ACE-I—angiotensin-converting enzyme inhibitor.

**Table 2 jcm-11-03994-t002:** Dynamics of LV indicators before CABG and after 18 months.

Characteristics	Group 1 Control (*n* = 36)	Group 2 before Surgery(*n* = 160)	Group 3 18 Months after CABG (*n* = 160)	H	*p*
Structural indicators and systolic function					
LA, ME [LQ; UQ] mm	4.15 [3.7; 4.55]	4.4 [4.1; 4.8] *	4.6 [4.2; 4.95] **#	15.98	0.0003
EDD, ME [LQ; UQ] mm	5.2 [4.95; 5.5]	5.5 [5.3; 6.1] *	5.5 [5.1; 6.0] #	18.3	0.0001
ESD LV, ME [LQ; UQ] mm	3.3 [3.05; 3.5]	3.6 [3.3; 4.0] *	3.7 [3.4; 4.2] #	23.04	<0.001
EDVI, ME [LQ;UQ] мл	130.0 [115.5; 147.0]	147.0 [135.0; 187.0] *	147.0 [124.0; 180.0] #	18.21	0.0001
ESV, ME [LQ;UQ] mL	44.0 [36.5; 51.0]	52.0 [44.0; 70.0] *	58.0 [46.0; 79.0] #	22.19	<0.001
LVEF, ME [LQ;UQ] %	62.0 [60.0; 64.0]	61.0 [55.5; 65.5]	56.0 [51.0; 61.0] **#	34.6	<0.001
LVM, ME [LQ;UQ] (g)	206.0 [172.0; 254.0]	303.5 [258.8; 360.4] *	242.0 [201.5; 286.5] **#	49.65	<0.001
LVMI, ME [LQ; UQ]	101.5 [89.0; 136.0]	136.2 [87.0; 173.4]	138.0 [112.0; 159.0]	0.85	0.654
Indicators of LV diastolic function					
IVRT, ME[LQ;UQ] m/s	90.0 [90.0; 90.0]	92.0 [90.0; 98.0] *	90.0 [88.0; 92.0] #	46.8	<0.001
E, ME [LQ; UQ] cм/ceк	56.5 [48.0; 65.0]	58.0 [46.0; 68.0]	56.0 [47.0; 65.0]	0.73	0.691
A, ME [LQ; UQ] cm/s	66.0 [56.0; 74.0]	67.0 [58.5; 78.0]	63.0 [54.0; 71.5]	7.93	0.189
E/A	0.83 [0.69; 1.2]	0.79 [0.68; 1.1]	0.81 [0.7; 1.24]	2.28	0.319
e’, ME [LQ; UQ] cm/s	9.4 [8.5; 11.0]	9.3 [7.5; 11.0]	10.6 [9.0; 12.3] **#	16.39	0.0003
a’, ME [LQ;UQ] cm/s	11.95 [10.5; 13.3]	10.0 [8.6; 11.6] *	9.4 [7.7; 11.0] **#	21.07	<0.001
e’/a’, ME [LQ;UQ]	0.77 [0.66; 1.14]	0.91 [0.68; 1.29]	1.24 [0.83; 1.48] **#	25.66	<0.001
s’, ME [LQ; UQ] cm/s	10.0 [9.5; 10.7]	9.2 [8.2; 10.35] *	9.8 [8.6; 10.8] **	14.18	0.0008
E/e’, ME [LQ;UQ]	5.9 [5.13; 6.73]	6.1 [4.75; 7.6]	5.1 [4.17; 6.38] **#	15.82	0.0004
Tei LV [LQ;UQ]	0.29 [0.26; 0.34]	0.33 [0.25; 0.43] *	0.26 [0.22; 0.31] **#	37.47	<0.001

Continuous data are presented as the median [lower quartile, upper quartile]. CABG—coronary artery bypass grafting; EDD—end-diastolic dimension; EDVI—end-diastolic volume index; EF—ejection fraction; LA—left atrium; LVMI—left ventricular mass index; ESV—end-systolic volume; ESD—end-systolic dimension; IVRT—isovolumic relaxation time; E—peak early diastolic left ventricular filling velocity; A—peak left ventricular filling velocity at atrial contraction; E/A—ratio of peak early diastolic filling velocity to peak filling velocity at atrial contract; e’—early diastolic mitral annular tissue velocity; a’—late diastolic mitral annular tissue velocity; e’/a’—ratio of the velocities of early and late movements of the mitral annulus; s’—systolic mitral annular tissue velocity; E/e’—ratio of the early diastolic velocity of mitral inflow to the early diastolic velocity of mitral annular motion; Tei LV—myocardial performance index left ventricular; * *p* < 0.05 between groups 1 and 2; ** *p* < 0.05 between groups 1 and 3; # *p* < 0.05 between groups 2 and 3.

**Table 3 jcm-11-03994-t003:** Dynamics of the lifespan indicators before CABG and after 18 months.

Characteristics	Group 1 Control(*n* = 36)	Group 2 before Surgery(*n* = 160)	Group 3 18 Months after CABG (*n* = 160)	H	*p*
Structural indicators and systolic function					
RV, [LQ; UQ] mm	2.0 [1.8; 2.4]	2.0 [1.9; 2.2]	2.1 [2.0; 2.3]	2.39	0.302
RVth, [LQ; UQ] mm	0.3 [0.3; 0.3]	0.4 [0.3; 0.4] *	0.4 [0.3; 0.4] #	20.5	<0.001
TAPSE, [LQ; UQ] mm	26.0 [24.0; 28.0]	23.0 [21.0; 26.0] *	20.0 [18.0; 21.0] **#	107	<0.001
RVEF, ME [LQ; UQ] %	55.0 [54.0; 59.5]	55.0 [53.0; 57.0]	54.5 [52.0; 55.0] **#	15.0	0.0006
RA, ME [LQ; UQ] mm	44.0 [33.0; 55.0]	40.0 [33.0; 49.0]	53.0 [43.0; 63.0] **#	53.9	<0.001
mPAP. [LQ; UQ] mmhg	11.0 [11.0; 11.5]	12.0 [24.0; 29.0] *	11.0 [11.0; 13.0] #	9.6	0.008
sPAP. [LQ; UQ] mmhg	22.5 [21.0; 24.0]	27.0 [24.0; 29.0] *	24.0 [23.0; 28.0] #	9.81	0.007
Diastolic function indicators					
Et, ME [LQ; UQ] cm/s	50.0 [43.0; 55.5]	44.0 [37.0; 49.5] *	49.5 [44.0; 58.0] **	27.2	<0.001
At, ME [LQ; UQ] cm/s	37.0 [33.0; 45.0]	43.0 [35.0; 49.5] *	34.5 [30.0; 41.5] **	35.7	<0.001
E_t_/A_t_	1.41 [1.12; 1.53]	1.05 [0.76; 1.36] *	1.46 [1.29; 1.62] **	61.1	<0.001
e’t, ME [LQ; UQ] cm/s	10.7 [9.0; 12.5]	9.6 [8.05; 11.3] *	8.0 [7.0; 9.9] **#	54.1	<0.001
a’t, ME [LQ; UQ] cm/s	15.4 [12.85; 18.0]	14.0 [12.1; 15.8] *	9.4 [8.0; 11.0] **#	151	<0.001
e’t/a’t, ME [LQ; UQ]	0.7 [0.63; 0.78]	0.69 [0.59; 0.8]	0.78 [0.66; 1.19] **#	33.7	<0.001
s’t, ME [LQ; UQ] cm/s	14.0 [12.15; 15.9]	13.2 [11.9; 14.7]	11.8 [10.4; 12.95] **#	48.4	<0.001
Et/e’t, ME [LQ; UQ]	4.76 [3.58; 5.31]	4.47 [3.6; 5.5]	6.0 [4.77; 7.22] **#	64.2	<0.001
Right ventricular Tei index [LQ; UQ]	0.31 [0.26; 0.33]	0.3 [0.24; 0.38]	0.27 [0.21; 0.32] **#	17.6	0.0001

Continuous data are presented as the median [lower quartile, upper quartile]. CABG—coronary artery bypass grafting; RV—right ventricular; mPAP—mean pulmonary arterial pressure; sPAP—systolic pulmonary arterial pressure; TAPSE—tricuspid annular plane systolic excursion; Tei—myocardial performance index; RVth—thickness of right ventricular wall in diastole; EF—ejection fraction; RA—right atrium; Et—early transtricuspid diastolic filling; At—late transtricuspid mitral diastolic filling; e’t—early diastolic tricuspid annular tissue velocity; a’t—late diastolic tricuspid annular tissue velocity; e’t/a’t—ratio of early diastolic tricuspid annular tissue velocity to the late diastolic tricuspid annular tissue velocity; s’t—systolic tricuspid annular tissue velocity; Et/e’t—ratio of early transmitral diastolic filling to the early diastolic tricuspid annular tissue velocity; * *p* < 0.05 between groups 1 and 2; ** *p* < 0.05 between groups 1 and 3; # *p* < 0.05 between groups 2 and 3.

**Table 4 jcm-11-03994-t004:** Association of variables before surgery with RVSD presence 18 months after CABG (binary logistic regression analysis, forward likelihood ratio).

Variables in the Equation							
		B	SE	Wald	df	Significance	Exp(B)
Step 1 ^a^	e’t	−0.212	0.089	5.686	1	0.017	0.809
	Constant	1.318	0.875	2.266	1	0.132	3.736
Step 2 ^b^	TAPSE	0.117	0.052	5.055	1	0.025	1.124
	e’t	−0.302	0.102	8.717	1	0.003	0.740
	Constant	−0.547	1.212	0.203	1	0.652	0.579
Step 3 ^c^	TAPSE	0.147	0.054	7.368	1	0.007	1.158
	e’t	−0.293	0.104	7.997	1	0.005	0.746
	RVSD	1.150	0.505	5.192	1	0.023	3.158
	Constant	−1.529	1.297	1.390	1	0.238	0.217

^a^ Variable(s) entered during step 1: e’t. ^b^ Variable(s) entered during step 2: TAPSE. ^c^ Variable(s) entered during step 3: RVSD—right ventricular systolic dysfunction before surgery; CABG—coronary artery bypass grafting; e’t—velocity of early diastolic movement of the tricuspid valve annulus; TAPSE—tricuspid annular plane systolic excursion before surgery.

**Table 5 jcm-11-03994-t005:** Association of variables before surgery with RVDD presence 18 months after CABG (binary logistic regression analysis, forward likelihood ratio).

Variables in the Equation							
		B	SE	Wald	df	Significance	Exp(B)
Step 1 ^a^	At	0.036	0.016	4.995	1	0.025	1.037
	Constant	−1.213	0.707	2.943	1	0.086	0.297
Step 2 ^b^	Myocardial infarction history	0.809	0.357	5.132	1	0.023	0.445
	At	0.038	0.017	5.327	1	0.021	1.039
	Constant	−0.798	0.744	1.151	1	0.283	0.450

^a^ Variable(s) entered during step 1: At. ^b^ Variable(s) entered during step 2: myocardial infarction history; RVDD—right ventricular diastolic dysfunction before surgery; CABG—coronary artery bypass grafting; At—the velocity of late transtricuspid filling.

**Table 6 jcm-11-03994-t006:** Linear regression analysis (stepwise method) for the relationship of s’t 18 months after CABG with echocardiography and clinical variables before surgery and perioperative complications.

Coefficients ^a^						
Model		Unstandardized Coefficients		Standardized Coefficients	*t*	Significance
		B	SE	Beta		
1	(Constant)	9.445	0.725		13.033	0.000
	e’_t_	0.231	0.070	0.257	3.301	0.001
2	(Constant)	8.139	0.924		8.811	0.000
	e’t	0.178	0.073	0.198	2.435	0.016
	s’t	0.138	0.061	0.182	2.237	0.027

^a^ Dependent Variable: s’t 18 months after CABG. The original model included: CABG—coronary artery bypass grafting; e’t—velocity of early diastolic movement of the tricuspid valve annulus; s’t—velocity of systolic movement of the tricuspid valve annulus.

**Table 7 jcm-11-03994-t007:** Linear regression analysis (stepwise method) for the relationship of Et/e’_t_ 18 months after CABG with echocardiography and clinical variables before surgery and perioperative complications.

Model		Unstandardized Coefficients		Standardized Coefficients	*t*	Significance
		B	SE	Beta		
1	(Constant)	2.080	1.532		1.357	0.177
	LA	0.926	0.344	0.212	2.694	0.008
2	(Constant)	−1.671	2.077		−0.804	0.422
	LA	0.902	0.338	0.207	2.672	0.008
	age	0.061	0.023	0.202	2.618	0.010
3	(Constant)	−2.953	2.157		−1.369	0.173
	LA	0.936	0.335	0.214	2.796	0.006
	age	0.057	0.023	0.191	2.483	0.014
	Et	0.030	0.015	0.152	1.980	0.050
4	(Constant)	−3.331	2.143		−1.554	0.122
	LA	0.986	0.332	0.226	2.967	0.004
	age	0.058	0.023	0.194	2.550	0.012
	Et	0.031	0.015	0.157	2.065	0.041
	Stroke in the early postoperative period	1.692	0.831	0.155	2.037	0.043

Dependent variable: Et/e’t 18 months after CABG. The original model included: CABG—coronary artery bypass grafting; LA—left atrium dimension; Et—the velocity of early transtricuspid filling.

## Data Availability

The datasets used and/or analyzed during the current study available from the corresponding author on reasonable request.

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
