# Peer review of "Indicators of the Right Ventricle Systolic and Diastolic Function 18 Months after Coronary Bypass Surgery"

_jcm, 2022, doi:10.3390/jcm11143994_

Round 1

Reviewer 1 Report

I have a few comments about this study.

  1. I think the authors should describe the intraoperative data (rate of off-pump procedure, number of distal anastomosis, operation time as well as cardiopulmonary bypass and aortic cross-clamp time).

  1. In line 209 to 210, the authors stated “… and the highest values of TAPSE (P<0.001)”. I found that the CAD group after CABG had the lowest value of TAPSE in Table 3.

  1. Do the authors have the early postoperative echocardiographic data of CAD patients? I am interested in the proportion of patients having RVSD or RVDD in the early postoperative period and the comparison of early postoperative and mid-term incidence of RV dysfunction.

  1. The present study demonstrated that both systolic and diastolic RV function had deteriorated in patients with CAD 18months after CABG. As the authors pointed out, decreased RV function can be associated with worse clinical outcomes. Do they consider that CABG worsened the RV function in CAD patients? I would like to know their opinion about perspectives of surgical coronary revascularization in patients with CAD, especially in those with preoperative RV dysfunction.

Author Response

First of all, I would like to thank the reviewer for the great work done in evaluating our manuscript and for the useful comments that allowed us to improve the manuscript. Responses to the comments of the reviewer:

  1. I think the authors should describe the intraoperative data (rate of off-pump procedure, number of distal anastomosis, operation time as well as cardiopulmonary bypass and aortic cross-clamp time).

Reply: Thanks for the suggestion; we have added intraoperative data to table 1

  1. In line 209 to 210, the authors stated “… and the highest values of TAPSE (P<0.001)”. I found that the CAD group after CABG had the lowest value of TAPSE in Table 3.

Reply: Thanks for the note; we have fixed this technical error.

  1. Do the authors have the early postoperative echocardiographic data of CAD patients? I am interested in the proportion of patients having RVSD or RVDD in the early postoperative period and the comparison of early postoperative and mid-term incidence of RV dysfunction.

Reply: Without a doubt, the question of the proportion of patients with RVSD or RVDD in the early postoperative period and the comparison of early postoperative and mid-term rates of RV dysfunction is interesting. Unfortunately, the design of the study did not include an assessment of right ventricular function in the early postoperative period. We additionally noted this in the study limitations section.

  1. The present study demonstrated that both systolic and diastolic RV function had deteriorated in patients with CAD 18months after CABG. As the authors pointed out, decreased RV function can be associated with worse clinical outcomes. Do they consider that CABG worsened the RV function in CAD patients? I would like to know their opinion about perspectives of surgical coronary revascularization in patients with CAD, especially in those with preoperative RV dysfunction.

Reply: According to our results, as well as according to other studies, the fact of a decrease in right ventricular function after open heart surgery is beyond doubt. Immediately after the operation, his systolic dysfunction is detected in almost 80% of cases, after six months - in about 50%. According to our data, after 1.5 years - in 30% of cases. Apparently, this is a consequence of the impact of perioperative factors (sternotomy, pericardial incision, cardioplegia), and we have to come to terms with this. Yes, preoperative RV dysfunction is one of the factors for poor prognosis of CABG, which must be taken into account along with others (age of patients, recent myocardial infarction, comorbid conditions, etc.). It is possible that when determining the tactics of myocardial revascularization in such cases, the choice should be made in favor of less invasive methods.

Reviewer 2 Report

Thank you for inviting me to review this manuscript. It is a clinical research including longterm follow-up data of perioperative right ventricular dysfuntion after CABG. The follow-up rate is relatively high. The study protocol was well written. The scientific meaning is very clear. However, the evaluation must be optimized. Some further statistics evaluation must be made. The rate of postoperative RV dysfunction is relatively high. 

After careful reading of the manuscript, some following considerations can be made: 

Comment 1: Please define “CAD” before you use.

Comment 2: Line 26-27: I cannot understand what you mean. Please reconsider the sentence.

Comment 3: please unify the decimal point “28.7+-3.69”->”28.7+-3.7”; “16.25%”->”16.3%”; “8.13%”->”8.2%”; “ 1.25%”->”1.3%”; “3.13%”->”3.1%”

Comment 4: Typo? Line 174 “1. Carotid artery stenosis”?

Comment 5: Table 1: how many patients had relevant stenosis in right coronary system? It would be important for perioperative RVSD/RVDD event assessment. 

Comment 6: as you show in this manuscript, about half of the whole patients RV Function in CABG group became worse than the baseline function. This outcome might lead the wrong message to the reader, that CABG makes RV function worse or your center operative results is bad. Please reconsider this point. Maybe you could evaluate more precisely what kind of patients-group had those event, for example, incomplete revascuralisation in right coronary artery system and so on. 

Comment 7: please provide more perioperative information about revascularization including off-pump or on-pump, arterial revascuralisation, operator experience, number of bypass, target bypass vessels and rate of complete revascuralisation etc. Further, is there any postoperative myocard infarction? It might impact on postoperative RV dysfunction. 

Comment 8: please provide how postoperative RV dysfunction could be treated, could be prohibited etc. 

Author Response

First of all, I would like to thank the reviewer for the great work done in evaluating our manuscript and for the useful comments that allowed us to improve the manuscript.

Responses to the comments of the reviewer:

Comment 1: Please define “CAD” before you use.

Reply: Thank you for your comment. I added the abbreviation "CAD" after the first mention of coronary heart disease (line 18)

Comment 2: Line 26-27: I cannot understand what you mean. Please reconsider the sentence.

Reply: High values At before surgery (p=0.021) and old myocardial infarction (p=0.023) were significantly  associated with an increased likelihood of detection RV diastolic dysfunction 18 months after CABG (χ2(2) = 10.78, p =0.005).

Comment 3: please unify the decimal point “28.7+-3.69”->”28.7+-3.7”; “16.25%”->”16.3%”; “8.13%”->”8.2%”; “ 1.25%”->”1.3%”; “3.13%”->”3.1%”

Reply: Thank you for your comment. Corrections have been made to the text.

Comment 4: Typo? Line 174 “1. Carotid artery stenosis”?

Reply: Thanks for the note; we have fixed this technical error.

Comment 5: Table 1: how many patients had relevant stenosis in right coronary system? It would be important for perioperative RVSD/RVDD event assessment. 

Reply:

Comment 6: as you show in this manuscript, about half of the whole patients RV Function in CABG group became worse than the baseline function. This outcome might lead the wrong message to the reader, that CABG makes RV function worse or your center operative results is bad. Please reconsider this point. Maybe you could evaluate more precisely what kind of patients-group had those event, for example, incomplete revascuralisation in right coronary artery system and so on. 

Reply: Of course, we are also saddened by such a frequency of persistence of RV dysfunction 18 months after surgery. However, the fact that the RV function decreases immediately after CABG is known, there are studies on this topic in other centers (for example, Korshin A, et al, 2019, doi: 10.1016/j.jtcvs.2018.09.114, Hashemi N, et al, 2018, doi: 10.1093/icvts/ivx420), this is not a feature of our center. Therefore, our data cannot be unexpected for the reader. Another thing is that we have shown that such dysfunction persists even 1.5 years after the operation, this fact deserves further study. For example, the clinical and prognostic significance of the presence of systolic/diastolic RV dysfunction at this postoperative time frame remains to be assessed. Our attempts to identify any perioperative factors influencing the presence of RV dysfunction in the middle-term after CABG have not been successful (see the answer to the next question).

Comment 7: please provide more perioperative information about revascularization including off-pump or on-pump, arterial revascuralisation, operator experience, number of bypass, target bypass vessels and rate of complete revascuralisation etc. Further, is there any postoperative myocard infarction? It might impact on postoperative RV dysfunction. 

Reply: Thank you for your comment. We added information about the perioperative period to Table 1 (CABG off-pump or on-pump, number of bypass grafts, duration of aortic cross-clamping, duration of CPB, combined interventions, incidence of right coronary artery disease). I would also like to clarify that during the operations, complete myocardial revascularization was achieved, the mammary artery was used for LAD revascularization, and the vein grafts were used for the remaining arteries. Patients were operated on by experienced teams of surgeons with more than 20 years of experience. The number of postoperative infarctions was small - only 2 (indicated in Table 1), respectively, they could not influence the development of subsequent RV dysfunction in such a number of patients. Perioperative parameters in univariate logistic regression analysis did not affect the presence of systolic and diastolic RV dysfunction 18 months after CABG (see additional tables in the attachment), so we did not include them in the multiple regression models.

Comment 8: please provide how postoperative RV dysfunction could be treated, could be prohibited etc.

Reply: As far as I understand, there are no specific treatments for postoperative right ventricular dysfunction. One of the reasons for this is the fact that in a large number of patients after surgery, over time (six months to a year), the indicators of systolic function of the right ventricle normalize. On the other hand, the clinical and prognostic significance of the preservation of right ventricular dysfunction after a year of observation remains unclear. Also, CABG without CPB was accompanied by a decrease in RV function to the same extent as on-pump operations (Pegg TJ, et al, 2008; doi: 10.1161/CIRCULATIONAHA.107.735621).

Round 2

Reviewer 1 Report

Appropriately replied.

Author Response

We would like to thank the reviewer for taking the time to evaluate our article and for highly appreciate the work we have done to correct the comments.

Reviewer 2 Report

in spite of the revision process the quality of the publication was not improved.

Author Response

We are very sorry that our arguments in response to the comments of the reviewer did not seem convincing to him. Perhaps the reason for this was technical errors committed by us in response to previous comments.
Firstly, we did not provide a response to the 5th reviewer's comment, and secondly, we did not attach an additional table with one-way logistic regression analysis calculations confirming our data and responses to the comments. We fix our mistakes. We hope this will help the reviewer in further evaluation of our manuscript.
